# NKG2A Immune Checkpoint in Vδ2 T Cells: Emerging Application in Cancer Immunotherapy

**DOI:** 10.3390/cancers15041264

**Published:** 2023-02-16

**Authors:** Valentina Cazzetta, Delphine Depierreux, Francesco Colucci, Joanna Mikulak, Domenico Mavilio

**Affiliations:** 1Laboratory of Clinical and Experimental Immunology, IRCCS Humanitas Research Hospital, 20089 Rozzano, Italy; 2Department of Medical Biotechnology and Translational Medicine, University of Milan, 20129 Milan, Italy; 3Department of Obstetrics & Gynaecology, University of Cambridge, National Institute for Health Research Cambridge Biomedical Research Centre, Cambridge CB2 0SW, UK; 4University of Cambridge Centre for Trophoblast Research, Cambridge CB2 0SW, UK; 5Division of Human Biology, Fred Hutchinson Cancer Center, Seattle, WA 98109, USA

**Keywords:** γδ T cells, NKG2A, inhibitory receptors, immune checkpoint inhibitors, cancer immunotherapy, therapeutic monoclonal antibodies

## Abstract

**Simple Summary:**

Boosting effector T cell anti-tumor response remains a challenge, in part owing to the expression of immune checkpoints and their ligands, such as NKG2A and HLA-E. Targeting NKG2A by gene knockout or blocking antibodies improves the cytotoxicity of Vδ2 T cells, a specific subset of human unconventional γδ T lymphocytes. Thus, a suitable selection of NKG2A^+^ or NKG2A^−^ Vδ2 T cells for expansion or engineering could help to narrow the Vδ2 T cell population according to the expression of HLA-E on tumor cells. With this emerging knowledge, approaches to target NKG2A in Vδ2 T cells might be a promising step forward to boosting Vδ2 T cell-based cancer immunotherapies.

**Abstract:**

Immune regulation has revolutionized cancer treatment with the introduction of T-cell-targeted immune checkpoint inhibitors (ICIs). This successful immunotherapy has led to a more complete view of cancer that now considers not only the cancer cells to be targeted and destroyed but also the immune environment of the cancer cells. Current challenges associated with the enhancement of ICI effects are increasing the fraction of responding patients through personalized combinations of multiple ICIs and overcoming acquired resistance. This requires a complete overview of the anti-tumor immune response, which depends on a complex interplay between innate and adaptive immune cells with the tumor microenvironment. The NKG2A was revealed to be a key immune checkpoint for both Natural Killer (NK) cells and T cells. Monalizumab, a humanized anti-NKG2A antibody, enhances NK cell activity against various tumor cells and rescues CD8 αβ T cell function in combination with PD-1/PD-L1 blockade. In this review, we discuss the potential for targeting NKG2A expressed on tumor-sensing human γδ T cells, mostly on the specific Vδ2 T cell subset, in order to emphasize its importance and potential in the development of new ICI-based therapeutic approaches.

## 1. Introduction

Unconventional gamma delta (γδ) T cells are considered a bridge between adaptive and innate immunity that undergo somatic gene rearrangement for the generation of T cell receptor (TCR) γ-chain in combination with a TCR δ-chain. Differently from conventional alpha beta (αβ) T cells, these cells are characterized by their different mechanisms and capabilities to recognize antigens without the major histocompatibility complex (MHC) restriction [1]. The ontology of γδ T cells occurs in the thymus, where the generation of a functional γδ TCR takes place in different stages of development [2]. Subsequently, these cells migrate to the peripheral blood and tissues, where they are involved in tissue homeostasis and immunosurveillance [3]. γδ T cells are known to act as stress sensors that exhibit effective immune responses against tumor-transformed cells and pathogen infections, such as Mycobacterium tuberculosis, Listeria monocytogenes, influenza viruses, Human Immunodeficiency virus (HIV), Epstein–Barr virus (EBV), and Hepatitis B virus (HBV) [4,5]. γδ T cells have broad functional properties, including cytotoxicity and the secretion of soluble factors such as cytokines (i.e., interferon-γ (IFN-γ), tumor necrosis factor-α (TNF-α)), interleukin (IL)-4, IL-10, IL-17, chemokines (i.e., Chemokine (C-C motif) ligand 3 (CCL3) and CCL4), and growth factors, including keratinocyte growth factor (KGF) and transforming growth factor-β (TGF-β), whose release depend on the types of signals present in the tissue microenvironment [6,7,8]. Additionally, γδ T cells can act as professional antigen-presenting cells and regulate the maturation of dendritic cells and B cell antibody (Ab) production [9,10,11]. In humans, γδ T cells are divided into two main subsets based on the Vδ chain used to make their TCR. In adults, Vδ1 T cells are predominantly found in peripheral tissues, such as the gut, liver, thymus, and skin; however, little is known about their antigens [12]. Vδ2 T cells are the prevalent γδ T lymphocytes in the peripheral blood, where they comprise 80–85% of all peripheral γδ T cells. TCR Vδ2 chains are frequently paired with Vγ9 chains during development and recognize phosphoantigens (PhAgs), small non-peptidic phosphorylated molecules that are intracellular metabolites of the mevalonate pathway [13,14]. In addition, human Vδ3 cells make up a minor γδ T cell subset. Vδ3 T cells are enriched in the liver and in patients with some chronic viral infections and leukemias. They bind to glycolipid antigens presented by the MHC class I-like molecule CD1d and annexin A2 [15,16,17,18]. Other human γδ T cell subsets, such as Vδ4, Vδ5, Vδ6, Vδ7, and Vδ8 T cells, have been mainly found in patients with pathological conditions [19,20]. The main features of human γδ T cell subsets in relation to conventional αβ T cells are summarized in Table 1.

NKG2A is a member of the C-type lectin receptors for MHC class I and is encoded by the gene complex located within the Natural Killer (NK) complex (NKC) on human chromosome 12p12-13. This gene complex varies in gene content between species and can encode both activating and inhibitory polymorphic receptors [21]. In humans, KLRC1 is one of four functional killer cell lectin-like receptor KLRC genes (NKG2A, C, E, and F) that, together with KLRD1 (CD94), form a heterodimeric NKG2A/CD94 inhibitory receptor (Figure 1A) [22]. NKG2A is a single-pass type II integral membrane glycoprotein that contains cytoplasmic, transmembrane, and extracellular lectin-like domains [23]. The intracellular portion has two immunoreceptor tyrosine-based inhibition motifs (ITIMs), whose phosphorylation recruits the intracellular phosphatases SHP-1 and -2 and induces an inhibitory signal upon binding to its ligand, the non-classical class I human leukocyte antigen E (HLA-E) [24,25,26,27]. The expression of NKG2A was detected in cytotoxic lymphocytes, including most NK cells, type 1 innate lymphocytes (ILC1), unconventional NKT cells, different subsets of CD8 αβ T and γδ T cells, specifically on Vδ2 T cell subset, in which its inhibitory signal suppresses their activation and its blocking can effectively unleash their effector response [28,29,30,31,32,33].

HLA-E, the ligand of NKG2A, is lowly expressed on almost all cell surfaces in human tissues, displays limited polymorphism, and presents peptides derived from the leader sequences of the classical MHC class I molecules HLA-A, HLA-B, and HLA-C [34,35]. Thus, the expression of HLA-E is directly related to the number of HLA-I molecules in a given cell, and NKG2A acts as a sensor to assess the net overall expression of HLA-I molecules on a target cell. Importantly, the expression of HLA-E greatly increases in several human malignancies with consequent inhibition of the effector functions of tumor-infiltrating lymphocytes (TILs) through its binding with NKG2A [36]. This interaction between NKG2A and HLA-E contributes to tumor immune escape; therefore, NKG2A-mediated mechanisms are currently being exploited to develop potential anti-tumor therapeutic strategies [30,34,37].

In this review, we summarize and discuss the current state-of-the-art knowledge regarding the potential of human Vδ2 T cells to be used in cancer immunotherapy with a focus on the NKG2A immune checkpoint (IC) in order to emphasize its importance and potential in the development of new IC-based therapeutic approaches in the near future.

## 2. Anti-Tumor Potential of γδ T Cells

There is emerging evidence that γδ T cells exhibit persistent cancer immunosurveillance. In a transcriptomic analysis of 18,000 tumor samples using the CIBERSORT algorithm [38], an abundance of γδ T cells was shown to be the most favorable prognostic parameter across 39 different human cancers, supporting the notion that γδ T cells are critical for optimal tumor defense. Recently, the improved computational CIBERSORT analysis confirmed the positive correlation between the abundance of Vδ2 TILs and favorable outcomes of patients with cancers such as colorectal carcinoma (CRC), prostate carcinoma, chronic lymphocytic leukemia (CLL), and acute myeloid leukemia (AML) [39]. Extensive functional characterization of total γδ TILs, as well as different γδ T cell subsets, has also been performed in vitro (e.g., melanoma, glioblastoma, renal, breast, lung, ovarian, colon, pancreatic), demonstrating their efficient capability to kill primary tumor cells and tumor cell lines isolated form melanoma, glioblastoma (GBM), renal, breast, lung, ovarian, colon, and pancreatic tumors [5,40]. In addition to the transcriptomic and functional in vitro characterization of γδ T cells, numerical and phenotypic analyses have been conducted in relation to clinical outcomes. Such investigations have correlated γδ T cells with better clinical outcomes for different malignancies, including melanoma, gastric cancer, primary CRC and liver metastatic colorectal carcinoma (CRLM), hepatocellular carcinoma (HCC), and others [17,41,42,43,44,45,46,47]. On the other hand, γδ TILs have been shown to have a pro-tumorigenic role that stems from their ability to produce IL-17 [8]. This relevance mainly originated in murine models and can be questioned by their conspicuous absence in human γδ T cells that require a highly inflammatory milieu to produce IL-17 that is difficult to reproduce in vitro [29,48]. Indeed, IL-17 can be secreted by Vδ2 T cells upon stimulation with cytokines such as IL-1β, IL-6, IL-23, and TGF-β [48,49]. Moreover, in humans, the tissue-resident Vδ1 T cells seem to be more prone to producing IL-17 than circulating Vδ2 T cells [50,51,52]. IL-17-producing by total γδ T cells have been observed in patients affected by CRC, cervical cancer, and others [51,52,53,54]. Other studies concluded that IL-17-producing γδ T cells in tumor samples, including ovarian cancer (OC) and CRLM, are negligible [17,46,51].

γδ T cells share a variety of anti-tumor mechanisms [55]. Commonly activated γδ T cells display strong cytotoxic activity through the release of granzymes and perforin and the production of IFN-γ and/or TNF-α cytokines to amplify the immune response and counteract tumor development [8,17,42]. Additionally, activating receptors expressed on different γδ T cell subsets include CD16, which binds to the Fc region of IgGs and induces antibody-dependent cellular cytotoxicity (ADCC) [8,17,56,57]. Moreover, both Vδ1 and Vδ2 T cells can kill their cancerous targets through signaling pathway activation of TNF family death receptors, such as FasL and TRAIL [58,59,60,61]. γδ T cells are uniquely equipped with two TCR-dependent and independent recognition pathways that are able to activate and sense γδ T cells against stressed and tumor cells. In fact, the recognition of tumor cells by γδ T cells has been attributed to the engagement of both the TCR and/or innate activating receptors, mostly NKG2D. NKG2D recognizes stress-inducible MHC class-I-related molecules (i.e., MHC class I-related chain A/B (MICA/B), UL16-binding proteins (ULBP1–6)), whose binding triggers the cytotoxic effector function of both Vδ1 and Vδ2 T cells independently of TCR signaling [62,63,64]. Moreover, natural cytotoxicity receptors (NCRs), i.e., NKp30, NKp44, NKp46, and DNAM-1 (CD226), can also be expressed on polyclonal γδ T cells and contribute to tumor cell recognition and killing [65,66]. In fact, NKp30 and NKp44 were detected in Vδ1 T cells [6,67]. Moreover, our recent study showed that NKp46 is specifically expressed in human intestine Vδ1 T intraepithelial lymphocytes (IELs) that are endowed with potent anti-tumor activity [42].

Specific TCR-dependent mechanisms that form the basis of γδ T cell-dependent anti-tumor responses are still poorly understood, mainly due to a lack of depiction of the complete repertoire of antigens recognized by γδ-TCRs and the specificity of different γδ T subsets [68]. The most well-known ligands for TCR-dependent activation were discovered in Vδ2 T cells, which bind to small, non-peptide PhAgs, such as isopentenyl pyrophosphate (IPP), secreted by tumor eucaryotic cells as a consequence of mevalonate pathway dysregulation [69,70]. A similar naturally occurring bacterial metabolite, hydroxyl dimethylallyl pyrophosphate (HDMAPP/HMBPP), is one of the strongest stimulants for Vδ2 T cells [71]. PhAgs are not necessarily displayed via MHC molecules. Instead, the butyrophilin (BTN) family members (BTNs), specifically BTN3A1 and BTN2A1, play crucial roles in PhAg sensing/presentation, activation, and the proliferation of Vδ2 T cells [72,73].

Overall, there is a substantial body of evidence supporting the notion that γδ T cells play important roles in tumor surveillance.

## 3. Vδ2 T Cells as a Source for Cancer Immunotherapies

Vδ2 T cells are attractive candidates for adoptive cell immunotherapy due to their unique features. The broad spectrum of cytotoxic activating receptors, as described above and expressed on these cells, are able to recognize cancer cells, reducing the chances of tumor immune escape through single antigen loss by αβ T cells. The innate-like features with MHC-independent activation easily allow Vδ2 T cell transfer between individuals in an allogenic setting without the risk of causing Graft versus Host Disease (GvHD) [74,75]. Moreover, their lower pattern of cytokine secretion, mainly IFN-γ and TNF-α, carries a lower risk of cytokine release syndrome. Furthermore, there is growing evidence to indicate that Vδ2 T cells can prime αβ T cell responses and interact with other immune cells, thereby enabling the orchestration of a cascade of immune responses against tumors [76]. Not only do unmodified Vδ2 T cells represent an attractive source for adoptive cell immunotherapy and for developing “off-the-shelf” Vδ2 T-based therapeutic products, but also genetic engineering strategies can be applied to further enhance the cytotoxicity and affinity of these cells towards specific tumor targets, either as vehicles for chimeric antigen receptors (CARs) or αβ T cell-derived TCRs or as bispecific T cell engagers (BiTEs). These specific applications have all been extensively described in several recent reviews [77,78,79,80,81,82,83,84].

The main obstacle against the application of γδ T cells for cell immunotherapy remains their effective in vitro or in vivo expansion while maintaining their high anti-tumor effector potential. Several protocols for the activation and expansion of both Vδ1 and Vδ2 T cells have recently been improved, thus allowing the acquisition of sufficient numbers of cells for utilization in the clinical setting [85,86]; however, Vδ2 T cells are preferential cancer immunotherapy targets due to their high concentration in the peripheral blood and their relatively simple activation and proliferation. Two main strategies allow us to harness Vδ2 T cell immunotherapeutic approaches based on their direct activation and expansion both in vivo and in vitro prior to adoptive transfer into patients. Indeed, pharmaceutical interventions with aminobisphosphonate drugs (ABPs) causing mevalonate pathway dysregulation, used in patients with excessive bone resorption, have been shown to cause systemic Vδ2 T cell stimulation and an increase in anti-tumor activity [87]. Following this observation, several clinical trials attempted to carry out in vivo stimulations of autologous Vδ2 T with ABPs (ABP pamidronate and zoledronate) and synthetic PhAg analogs (BrHPP and 2M3B1PP) alone or in combination with IL-2 in patients with multiple myeloma (MM), non-Hodgkin lymphoma (NHL), follicular lymphoma, AML, prostate cancer, renal cell carcinoma (RCC), CRC, breast cancer, melanoma, and neuroblastoma [88,89,90,91,92,93,94,95]. Although these approaches were well tolerated and no severe toxicity was observed, clinically variable antitumoral responses were reported. Similar protocolar approaches have been used for the ex vivo expansion of Vδ2 T cells in both autologous and allogenic settings and were reported to be safe but were inconsistent in terms of efficiency in the treatment of MM and leukemias (acute lymphoblastic leukemia (ALL), AML, and CLL) and different solid tumors such as RCC, CRC, non-small cell lung cancer (NSCLC), melanoma, and liver and gastric cancers [96,97,98,99,100,101,102,103,104]. However, in these studies, at least some patients showed a reduced tumor burden, regardless of their terminal disease. Thus, further studies, possibly conducted in patients at an earlier stage of disease, are necessary to explore the use of Vδ2 T cells as therapeutic targets. Moreover, a better understanding of the molecular mechanisms underlying Vδ2 T cell activation and the interaction of these cells with the tumor microenvironment (TME) is needed, thus highlighting the existence of new therapeutic molecules that can be targeted to improve immunotherapeutic approaches. In this context, the discovery of BTN as an immunoregulatory molecule involved in Vδ2 T cell activation led to the development of Imcheck ICT01, an activating humanized IgG1 Ab, which, upon binding to BTN3A, triggers Vδ2 T cell activation and increased cytotoxicity against BTN-expressing tumor cells. Encouraging activity of ICT01 was observed in the currently ongoing phase I/IIa clinical trial testing in monotherapy and in combined therapy with Pembrolizumab in patients affected by advanced-stage hematologic malignancies and in several solid tumors [105]. Although these promising results need to be further investigated, the fact that the drug combination therapy promises a better response will certainly advance our understanding of the role of immunotherapy in this setting.

In recent years, immunotherapies involving the blocking of ICs have achieved encouraging results for a variety of solid tumors. Thus far, only ICIs that target the PD-1 and CTLA-4 pathways have been approved for clinical use and include anti-CTLA-4 and anti-PD-1 Abs [106]. In particular, this therapeutic strategy has become the gold standard for melanoma treatment and may result in prolonged survival (2–3 years) for approximately 20–30% of patients [107]. Regarding the expression and functionality of ICs in the context of cancer, γδ T cells have been poorly investigated. The expression of ICs such as BTLA, TIGIT, PD-1, TIM-3, CD39, LAG-3, and NKG2A in γδ T cells has been observed in different solid and hematological tumors, including melanoma, neuroblastoma, CRC, breast cancer, OC, MM, AML, and lymphomas, and is associated with the aberrant activation and/or proliferation of γδ T cells [29,76,108,109] For instance, BTLA was found to be highly expressed by Vδ2 T cells in the lymph nodes of patients with lymphoma and could suppress their proliferation upon ligation by HVEM on primary tumors [110]. High PD-1 expression on γδ T cells has been reported in tumors such as neuroblastoma, CRC, and MM [111,112,113]. Interestingly, a recent study revealed that PD-1 blockade enhanced the ADCC of γδ T in an in vitro culture system [114]. However, controversial effects on Vδ2 T cell proliferation and cytotoxic activity have been reported in different human hematological malignancies [113,115]. In melanoma patients, higher proportions of γδ TILs expressing LAG-3 compared to control groups were associated with earlier relapse and shorter overall survival (OS) [116]. In AML and CRC patients, Vδ2 T cells displayed increased TIM-3 expression and a dysfunctional phenotype [117,118,119]. The activation of TIM-3 lowered Vδ2 cell cytotoxicity toward colon cancer cell lines [117]. In addition, Vδ2 cells from AML patients showed an impaired proliferative capacity, which was restored by blocking TIM-3 signaling [119]. When the expression levels of both TIM-3 and PD-1 were investigated, Vδ2 T cells co-expressing TIM-3 and PD-1 exhibited the lowest levels of IFN-γ and TNF-α production, and these increased upon anti-TIM-3 or anti-TIM-3 plus anti-PD-1 blocking of Abs but not with anti-PD-1 alone [118].

These data highlight the importance of targeting ICs for the functional restoration of γδ T cell activation in different malignancies.

## 4. Impact of NKG2A on the Effector Potential of Vδ2 T Cells

The earliest remark on the expression of NKG2A in human γδ T cells dates back to 1997 and shows that human γδ T cells, mostly Vδ2 T cells, harbor the CD94/NKG2A heterodimer [120,121]. Although this analysis was performed on a limited number of peripheral blood samples of healthy adults, our and other studies have confirmed these results by performing extensive multiparametric flow cytometry analyses on a vast number of analyzed samples [29,122]. NKG2A shows a high range of expression among Vδ2 T cells which, in healthy adults, covers from 20% to 90% of cells with a median of 50%, independently of age and gender. This high expression of NKG2A by Vδ2 T cells was further confirmed by transcriptional analysis [29,123,124]. On the other hand, low expression levels of NKG2A were observed in Vδ1 and Vδ3 T cells in the blood and in the liver tissue where these cells preferably reside [15,17,120,123,124]. Differently, NKG2C, which also binds HLA-E, although with a 6-fold lower affinity [125], is rarely expressed on Vδ2 T cells [29,122].

NKG2A plays a key role in the development of NK cells. In fact, our current knowledge on the development of NK cells states that NKG2A is the only HLA-specific inhibitory receptor expressed on the precursor CD56^bright^ cell subset of human NK cells, and its expression gradually diminishes on the mature CD56^dim^ NK cell subset [30,126]. Additionally, reconstitution of NK cells following allogenic hematopoietic stem cell transplantation (HSCT) demonstrated that circulating NK cells present a transient phenotype of CD56^dim^ cells expressing high levels of NKG2A that is progressively lost over time, thus implying NKG2A an important receptor for early stages of NK cell ontogenesis [127]. Instead, the role of NKG2A in the ontogeny of human γδ T cells is not defined yet. Interestingly, our and other studies indicate that the expression of this inhibitory receptor on Vδ2 T cells is already programmed in the postnatal pediatric thymus and can be detected in the blood soon after birth (within 10 weeks) [29,128,129]. On the other hand, neither the expression of NKG2A nor that of CD94 has been detected in cord blood [122,129]. The absence of CD94 in Vδ2 T cells in neonates and infants is not due to a generalized lack of CD94 expression, as NK cells express a level of CD94 that is comparable to that of adult NK cells [122]. This suggests that the acquisition of NKG2A could be triggered by the postnatal thymus environment and, once these cells are in the periphery, their concentration can expand in response to environmental and pathogenic microbes, resulting in the predominance of Vδ2 T cells in the majority of the adult population. Little is also known about the possible regulation of NKG2A expression in adults. The expression of NKG2A in Vδ2 T cells expanded in vitro does not increase upon PhAgs and IL-2 activation. Indeed, an unrelated in vitro expansion of NKG2A^+^ and NKG2A^−^ Vδ2 T cells was observed, suggesting the autonomous self-renewal of these two subpopulations [29].

In adults, NKG2A^+^ and NKG2A^−^ Vδ2 T cells show few phenotypical differences at both the protein and transcriptional levels, resulting in similarities in differentiation status and clonal expansion [29]. Moreover, differently from CD8 αβ T cells, in which the expression of NKG2A is induced during viral infections such as human cytomegalovirus (HCMV) [130], in γδ T cells, HCMV infection does not seem to impact the expression of NKG2A [29]. However, further studies are necessary to understand their possible roles in viral infections.

Importantly, minor phenotypic differences are offset by an evident functional divergency between NKG2A^+^ and NKG2A^−^ Vδ2 T cells [29,124]. Indeed, we and others have observed a paradigm whereby Vδ2 T cells harboring this negative receptor NKG2A are characterized by hyper-responsiveness [29,120]. Indeed, NKG2A^+^ Vδ2 T cells produce significantly higher amounts of IFN-γ and TNF-α upon PhAg TCR activation. In addition, NKG2A^+^ Vδ2 T cells present higher cytotoxic potential against HLA class I deficient tumor cell targets that do not express the NKG2A ligand, HLA-E. These data were also confirmed by transcriptional analysis at single-cell resolution, showing that Vδ2 T cells expressing NKG2A harbor a greater cytotoxic potential and ability to produce IFN-γ and TNF-α cytokines [29,48,122,123].

Interestingly, NKG2A^+^ Vδ2 T cells require higher amounts of PhAgs to produce their maximal cell response [120], thus indicating that the CD94/NKG2A complex does not prevent the activation of Vδ2 T cells by their ligands but rather affects their activation thresholds. In this context, there is growing evidence of the role of NKG2A in the so-called “education” of NK cells [131,132,133]. This process is finely tuned by the interactions of inhibitory receptors with their ligands to set the effector functions of NK cells at hyper-responsive levels in response to stimulatory activation (Figure 1B). In the case of γδ T cells, it is still not clear whether the hyper-responsive NKG2A^+^ Vδ2 T cells are “educated” or whether they represent more mature cells. In fact, the higher effector potential of NKG2A^+^ Vδ2 T cells could be a consequence of both their education and their maturation process. However, the presence of highly differentiated and clonally expanded NKG2A^+^ and NKG2A^−^ Vδ2 T cells in healthy adults indicates that the expression of NKG2A identifies a subset of Vδ2 T “educated” cells [29,48]. NK cell education primarily appears during cell development, although new findings suggest that this phenomenon also occurs under disease conditions [134]. The observed increase in the expression of NKG2A in postnatal γδ thymocytes indicates their possible educational process in the thymus early after birth [29,128,129]. These data are also in line with the observation that Vδ2 T cells acquire higher levels of cytotoxic mediators (e.g., granzymes, perforin, granulysin) in correlation with increased expression of NKG2A rapidly after birth [129]. In addition, it was observed that IL-23 drives in CD94^−^ Vδ2 T cells the acquisition of a cytotoxic program and the CD94^+^ phenotype [122].

The mechanisms that drive the education of NK cells through NKG2A are poorly understood, but it has been reported that HLA-E expression is fundamental. In particular, genetic modifications that influence HLA-E expression have shown a correlation between NKG2A expression and NK cell education. In fact, cell membrane expression of HLA-E requires the supply of peptides from classical HLA-A, HLA-B, or HLA-C for appropriate folding and transport to the cell surface [135]. There is a dimorphism of the leader sequence supplied by HLA-B encoding either threonine (T) or methionine (M) [136,137]. This separates individuals into those who can provide functional peptides for high HLA-E expression and NKG2A ligation, which leads to education and increased functional potency (MT or MM), and those who cannot (TT) and therefore have low HLA-E expression [138]. A role for this HLA-B dimorphism is emerging in patients with leukemia and GvHD [139,140]. Thus, it could be interesting to evaluate whether the functional potency of γδ T cells might be influenced by HLA-B alleles.

Overall, the expression of the NKG2A receptor defines distinct Vδ2 T cells with higher anti-tumor potential that may be useful for Vδ2 T-cell-based cancer immunotherapies. However, further studies are necessary to unveil the ability of NKG2A to enhance the cytotoxic potential of Vδ2 T cells. In this regard, several features of the human Vδ2 T cell compartment suggest similarities to mouse γδ T cell subsets [141]. Thus, it would be interesting to evaluate NKG2A expression in different subsets/effector-linage of murine γδ T cells. Indeed, recently the NKG2A knockout mouse model has been used to unveil the role of NKG2A in the education process of tissue-resident NK cells [133].

## 5. NKG2A Immune Checkpoint in Vδ2 T Cells

The hyper-responsiveness of NKG2A^+^ Vδ2 T cells can be counterbalanced by the inhibitory signaling of NKG2A upon binding with its HLA-E ligand, thus affirming that NKG2A is the crucial IC in Vδ2 T cells. Indeed, the CD94/NKG2A complex expressed on Vδ2 T cells is able to efficiently block the killing of tumor cells expressing HLA-E molecules [29,120,121,142]. In fact, masking with the monoclonal Ab (mAb) NKG2A/CD94 complex on Vδ2 T cells or MHC class I molecules on target cells increases the cytotoxic response of Vδ2 T cells to HLA-E-expressing tumor cells. Additionally, CRISPR/Cas9-induced NKG2A knockout in Vδ2 T cells activated and expanded in vitro enhances the killing of tumor cells, despite their HLA-E expression [29]. In addition, the cross-linking of NKG2A during CD3 stimulation reduces the cytotoxicity of Vδ2 T cells, confirming that NKG2A negatively regulates their cell function [122]. Although, in another study, no significant effect on the cytotoxic response of Vδ2 T cells was observed after CD94/NKG2A blocking with mAb. However, only two Vδ2 T cell clones, for which the expression level of NKG2A was not shown, were tested [143]. Overall, these results demonstrate that NKG2A is a functional IC in Vδ2 TILs and that the anti-tumor effector functions of NKG2A^+^ Vδ2 TILs are finely tuned by the degree of HLA-E expression on tumor target cells.

Interestingly, the CD94/NKG2A complex on Vδ2 T cells also regulates both lytic and proliferative activities against virus-infected cells [121]. Indeed, triggering the mAb CD94 inhibitory signal in Vδ2 T cells significantly reduces the lysis of HIV- and HBV-infected targets. Accordingly, the masking of HLA class I molecules promotes the Vδ2 cytotoxicity of virus-infected targets. Moreover, an in vitro expansion of peripheral blood Vδ2 T is reduced upon CD94 triggering, which can otherwise be induced by HIV-infected cells [121]. The NKG2A receptor, which is highly expressed in human small intestinal γδ T IELs, was also found to negatively regulate CD8 αβ T IEL activation in celiac disease [144]. This suppression occurred partially as a result of the engagement of NKG2A with its ligand, HLA-E, on enterocytes and/or αβ T IELs. This immunosuppressive effect can be reduced by blocking the NKG2A/HLA-E interaction. Importantly, patients with active celiac disease have significantly decreased frequencies of NKG2A^+^ γδ T IELs, thus indicating that the NKG2A receptor expressed on γδ T IELs acts as a key regulator of celiac disease.

Thus, physiologically, the expression of NKG2A on Vδ2 T interacts with ubiquitously expressed HLA-E, whose expression, although at low levels, on all nucleated cells may represent a mechanism to fine-tune functions of highly responsive NKG2A^+^ Vδ2 T cells [145,146]. On the other hand, the overexpression of HLA-E in pathological conditions such as solid and hematological malignancies may be responsible for tumor escape from Vδ2 T cell-mediated immunosurveillance [147]. It is also possible that the increase of soluble HLA-E plasma levels found in some tumors may be responsible for the inhibition of Vδ2 T cells [148,149].

Since overexpressed HLA-E can prevent tumor cell lysis mediated by Vδ2 T cells, the blockade of the NKG2A-HLA-E axis may enhance the Vδ2 T cell-based immunotherapeutic efficacy by administering anti-NKG2A/CD94 mAb to unleash Vδ2 T effector functions. Furthermore, the differences in the anti-tumor potential of NKG2A^+^ and NKG2A^−^ Vδ2 T provide functional indications of their potential uses and indicate the optimal choice according to the immunosuppressive HLA-E-mediated TME.

## 6. NKG2A in Cancer Immunotherapy—The Use of Vδ2 T Cells

The overexpression of HLA-E is predictive of poor prognostic outcomes in patients with different tumors (e.g., OC, GBM, CRC, RCC, and NSCLC) [150,151,152,153,154,155]. According to multiple studies, having an increased number of NKG2A^+^ TILs is correlated with a poor prognosis in patients affected by colorectal, gynecological, breast, and liver cancers [149,152,156,157,158,159]. Collectively, this evidence indicates that it is worth developing NKG2A-HLA-E axis blockade strategies for immunotherapy in cancer patients.

Monalizumab (IPH2201) is a humanized mAb that targets NKG2A [160]. When tested in preclinical studies, Monalizumab showed the capacity to enhance anti-tumor immunity by unleashing both NK cells and CD8 αβ T cells [161]. Andre et al. also showed that the combined blockade of NKG2A with Monalizumab in combination with PD-1/PD-L1 or Cetuximab, an anti-epidermal growth factor receptor (EGFR) mAb, enhanced anticancer immunity, suggesting that Monalizumab could be used in combination with other oncology treatments. In patients with pretreated gynecologic cancers, Monalizumab was found to be well tolerated and showed short-term disease stabilization [162]. Preliminary data from microsatellite stable (MSS)-CRC patients who do not typically respond to anti-PD-1/PD-L1-based therapy showed the clinical efficacy and safety of the combination of Monalizumab and Durvalumab (an anti-PD-L1 mAb) [160,161]. On the other hand, the clinical trial (NCT04590963) in patients affected by head and neck cancer (HNSCC) and receiving Monalizumab and Cetuximab as a combination therapy did not meet objective response (https://yhoo.it/3oOARub, accessed on 24 January 2023). Currently, there are a number of ongoing clinical trials testing the efficacy of Monalizumab alone or in combination with other ICIs for the treatment of hematological and solid tumors (Table 2) that will progress our knowledge.

The activity of the above therapies is mainly attributed to the NKG2A^+^ NK and conventional CD8 αβ T cells [161], and nothing is known about γδ T cells. Our study shows that Vδ2 T cells infiltrating tumors such as GBM, NSCLC, HCC, and CRLM express high levels of NKG2A [17,29]. In some tumors (e.g., GBM, NSCLC), the percentage of NKG2A^+^ Vδ2 T cells can exceed that of those circulating in the blood, although it is not clear whether this higher expression level is induced by the TME or whether it is due to the preferable infiltration of this subset to the tumor side. The degree of infiltration of highly cytotoxic NKG2A^+^ Vδ2 T cells in some tumors can exert different impacts on patients’ OS in relation to the degree of expression of HLA-E. High frequencies of NKG2A^+^ Vδ2 TILs significantly correlate with improvement in patients’ OS in NSCLC and HCC tumors with similar levels of HLA-E compared to that present in normal tissue. On the other hand, in cases of GBM, there is a higher expression of HLA-E coupled with the lack of any clinical impact of NKG2A^+^ Vδ2 T cells on patients’ OS. Although only correlative, these studies suggest that multivariate analyses matching the degree of HLA-E expression with the frequency of NKG2A^+^ Vδ2 T cells might identify reliable targets to enhance Vδ2 T cell activity in several tumors.

The implementation of Vδ2 T cells in different cancer therapeutic approaches, including their in vivo activation and expansion, adoptive cell transfer therapies, and genetic engineering, needs to optimize the efficacy of such treatments by selecting effector Vδ2 T cells that are endowed with the maximal anti-tumor potential (Figure 2). In this regard, we need to consider that hyper-responsive NKG2A^+^ Vδ2 T cells would provide a more powerful tool for eradicating malignant cells compared to adoptive cell transfer trials administering all γδ T cells. However, the clinical use of customized NKG2A^+^ or NKG2A^−^ Vδ2 T cells should be matched with the histopathologic features of HLA-E expression to tailor those immunotherapeutic protocols to exert the most powerful immune responses against cancers.

## 7. Conclusions and Future Directions

While immune checkpoint inhibitors targeting the PD1/PDL1 and CTLA-4 axes have revolutionized cancer treatment, novel drugs are needed to overcome the limitations of current adaptive ICI-based immunotherapies, including insufficient tumor antigens and poor antigen presentation. Therefore, additional mechanisms activating the innate and adaptive immune systems have the potential to increase anti-tumor responses and alleviate immunosuppression in the tumor microenvironment. In this regard, innate ICIs alone or in combination with adaptive ICIs are a promising novel anticancer strategy [163]. These drugs can unleash the innate immune system against tumors, generating natural cytotoxicity and synergistically activate the adaptive immune system. Unlike adaptive immunity, innate immunity provides immediate and broad immune responses, offering a quicker immune attack and access to a wide array of mechanisms to promote disease resolution. Moreover, there is considerable overlap in the expression of several ICIs between the innate and adaptive immune systems. In this context, NKG2A regulates both adaptive and innate immunity and could be an important candidate. NKG2A shows wide expression across cytotoxic lymphocytes, including innate NK cells, ILC1, unconventional NKT cells, and adaptive CD8 T lymphocytes. In addition, as we have summarized here, this inhibitory receptor is highly expressed by human Vδ2 T cells. Thus, generated Monalizumab, a blocking anti-human NKG2A mAb, has a great potential to liberate effector anti-tumor response of different cytotoxic lymphocytes. It is worth underlining that since NKG2A is specifically expressed on cytotoxic lymphocytes, compared with HLA-E, it is more suitable as the blockade target of the NKG2A-HLA-E axis. Indeed, HLA-E is expressed on almost all cell surfaces [34], thus making NKG2A blocking more specific.

The development of Vδ2 T-cell-based immunotherapy against malignant cells is a fast-evolving field. Due to the anti-tumor potential of these cells, several clinical trials are testing protocols that involve the activation and expansion of Vδ2 T cells, even those with variable efficacy. Therefore, the development of therapeutic strategies to boost the response of Vδ2 T cells through ICIs could allow the successful use of Vδ2 T cells as therapeutic agents in cancer. NKG2A blockade is a promising way to activate Vδ2 T cells. Several studies have highlighted the critical role of NKG2A in controlling the anti-tumor response of Vδ2 T cells. While clinical development is still needed, the targeting of NKG2A expressed on Vδ2 T cells has the potential to improve patient outcomes in many cancer types. However, there is still a vast amount of research that needs to be undertaken to select the most appropriate cancers to be treated with NKG2A blockers. Finally, though NKG2A blockade has shown limited effects as a stand-alone therapy, the NKG2A blocking mAb has synergistic effects with other tumor ICIs. Thus, synergistic effects to optimize the clinical benefits of NKG2A blockers in Vδ2 T cells could be realized by combining them with other onco-therapies such as ICT01, a BTN target, or other ICIs.

Despite Monolizumab, other anti-NKG2A mAbs are tested (KSQ). In particular, the use of novel screening techniques such as a semi-mechanistic pharmacokinetic/receptor occupancy (SM-PK/RO) model was used to identify anti-NKG2A mAbs with a high specific monovalent affinity for NKG2A and anti-tumor cytotoxicity [164]. This analysis suggests that the increased affinity of highly selective anti-NKG2A KSQ mAbs may translate into substantial clinical benefits by lowering the dose, and/or reducing the dosing frequency, while retaining the saturation of receptor occupancy in tumor tissue needed to achieve optimal therapeutic anti-tumor efficacy.

Thus, the implementation of NKG2A in the development of Vδ2 T-cell-based immunotherapy against tumors may further improve and expand their potency alone or in combination with other cancer treatment options.

## Figures and Tables

**Figure 1 cancers-15-01264-f001:**
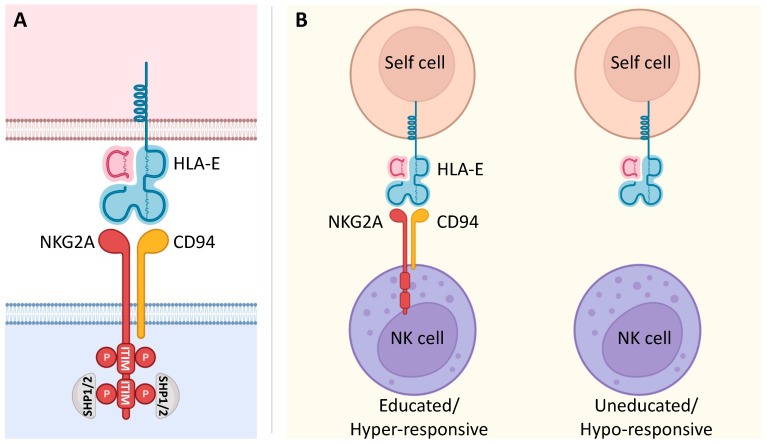
NKG2A signaling and its involvement in NK cell education. (**A**) The C-type lectin NKG2A receptor forms a heterodimer with CD94. It is characterized by a long cytoplasmic domain that contains two ITIMs. Upon binding to its ligand, inhibitory signaling of the non-classical MHC class I molecule HLA-E is based on the phosphorylation of two ITIMs with the consequent recruitment of the intracellular phosphatases SHP-1 and SHP-2. (**B**) NKG2A is known to be involved in NK cell education. In fact, the interaction of NKG2A with its ligand HLA-E renders NK cells educated and hyper-responsive, setting their effector functions at higher levels in response to stimulatory activation. On the other hand, NK cells that fail this interaction due to the lack of NKG2A are uneducated and hypo-responsive.

**Figure 2 cancers-15-01264-f002:**
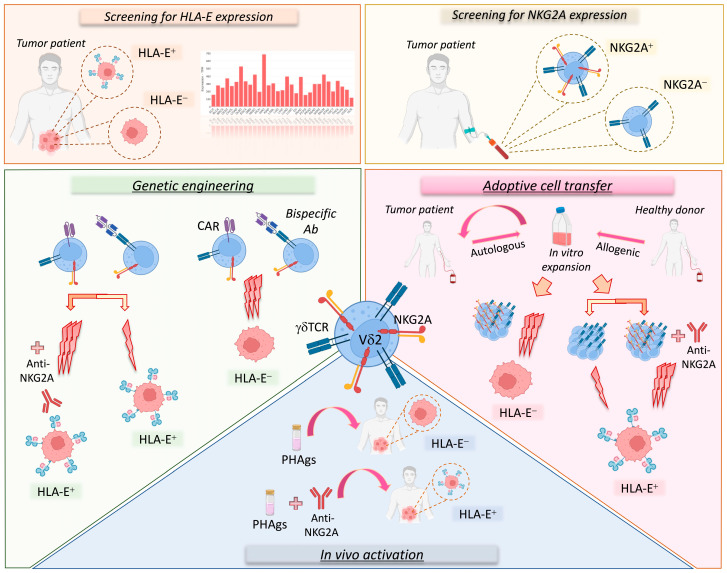
Vδ2 T cell-based immunotherapeutic approaches considering the NKG2A-HLA-E immune checkpoint axis. Several therapeutic strategies targeting the NKG2A-HLA-E axis on Vδ2 T cells could be used to treat cancer patients. First of all, it is important to evaluate the expression level of HLA-E in the tumor as well as that of NKG2A on Vδ2 T cells (upper panels). In fact, tumors can be characterized by variable levels of HLA-E (orange square; *HLA-E* transcription levels in various cancer types; RNA-sequencing data for the expression of *HLA-E* gene are reported as transcripts per million (TPM) and generated by using the Gene Expression Profiling Interactive Analysis 2 (GEPIA2) web server). Moreover, both patients and healthy donors can show variable levels of NKG2A expression on Vδ2 T cells (yellow square). These parameters, along with the possibility of sorting NKGA^+^ or NKG2A^−^ Vδ2 T cells, should be considered in immunotherapeutic strategies (lower panels). In particular, autologous or allogenic Vδ2 T cells can be expanded in vitro for adoptive cell transfer therapy. However, for tumors lacking HLA-E, it would be more suitable to use NKG2A^+^ Vδ2 T cells, as they show higher cytotoxic (3 lightning bolts) potential, while for tumors expressing HLA-E, it would be more appropriate to use NKG2A^+^ Vδ2 T cells in combination with anti-NKG2A mAbs or in alternative NKG2A^−^ Vδ2 T cells could be used with lower cytotoxic (1 lightning bolt) capability (pink area). Analogously, in genetic engineering approaches (green area) based on CAR-γδ T cells or bispecific antibodies, the use of NKG2A^+^ Vδ2 T cells against HLA-E^+^ tumors would be optimal only in the presence of the anti-NKG2A mAb. However, additional investigations are needed to establish the expression of NKG2A on engineered Vδ2 T cells. Finally, for the in vivo activation of Vδ2 T cells (blue square), PhAgs could be used alone or in combination with anti-NKG2A mAbs for cancer patients with low or high levels of HLA-E, respectively.

**Table 1 cancers-15-01264-t001:** Major differences among the main human γδ T and αβ T cell subpopulations.

Feature	γδ T Cells		αβ T Cells	
Subpopulations	Vδ1	Vδ2	CD4	CD8
Development	Thymus, common precursor			
TCR	TCRγδCD3ε_2_γδζ_2_		TCRαβCD3ε_2_γδζ_2_	
TCR repertoire	Variant Restricted number of γ and δ chains	Semi-invariant Vγ9Vδ2	Variant	
Frequency	25–40% peripheral tissues	1–10% PBMCs	25–60% PBMCs	5–30% PBMCs
MHC restriction	No MHC restriction		MHC class II	MHC class I
TCR ligands	Unprocessed unkown	Unprocessed PhAg	Processed peptides	
Response	Innate/Adaptive		Adaptive	

Abbreviations: TCR, T cell receptor; PBMCs, peripheral blood mononuclear cells; MHC, major histocompatibility complex; PhAg, phosphoantigen.

**Table 2 cancers-15-01264-t002:** Clinical trials with anti-NKG2A mAbs for the treatment of tumors.

Clinical Trial	Phase	Status	Drug	Disease
NCT04307329	II	Active, not recruiting	Monalizumab + Trastuzumab	Breast cancer
NCT04590963	III	Terminated	Monalizumab + Cetuximab	Squamous Cell Carcinoma of the Head and Neck
NCT05221840	III	Recruiting	Monalizumab + Durvalumab	Non-Small Cell Lung Cancer
NCT02921685	I	Unknown status	Monalizumab	Hematologic Malignancies
NCT02671435	I/II	Completed	Monalizumab + Durvalumab	Advanced Solid Tumors
NCT02557516	I/II	Terminated	Monalizumab + Ibrutinib	Chronic Lymphocytic Leukemia
NCT05414032	II	Not yet recruiting	Monalizumab + Cetuximab	Locoregionally Advanced Head and Neck Squamous Cell Carcinoma
NCT05061550	II	Recruiting	Monalizumab + Durvalumab	Non-Small Cell Lung Cancer
NCT02643550	I/II	Active, not recruiting	Monalizumab + Cetuximab + anti-PD-L1	Head and Neck Neoplasms
NCT04333914	II	Completed	Monalizumab	Advanced or Metastatic Hematological or Solid Tumor
NCT03822351	II	Active, not recruiting	Monalizumab + Durvalumab	Unresectable Stage III Non-Small Cell Lung Cancer
NCT03088059	II	Recruiting	Monalizumab	Recurrrent or Metastatic Squamous Cell Carcinoma of the Head and Neck
NCT03794544	II	Completed	Monalizumab + Durvalumab	Resectable Early-Stage Non-Small Cell Lung Cancer
NCT03833440	II	Recruiting	Monalizumab + Durvalumab	Advanced Non-Small Cell Lung Cancer
NCT05162755	I	Recruiting	S095029 ± Sym021 (anti-PD-1)	Advanced Solid Tumor Malignancies

Monalizumab = anti-NKG2A, Trastuzumab = anti-HER2, Cetuximab = anti-EGFR, Durvalumab = anti-PD-L1, Ibrutinib = anti-CD20.

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
