# Peer review of "NKG2A Immune Checkpoint in Vδ2 T Cells: Emerging Application in Cancer Immunotherapy"

_cancers, 2023, doi:10.3390/cancers15041264_

Round 1

Reviewer 1 Report

I have embedded my 6 comments in the attached pdf of the manuscript.

The recent setback for monalizumab + cetuximab in HNSCC should be considered.

One of the items that is always challenging when reading gamma delta manuscripts is the use of "gd" that does not specify whether the authors are describing d1 or d2 or both, which for the purposes of this review should ideally be focused around d2. Identifying this subset in the text whenever possible or indicating that the statement applies to d1/d2, rather than gd, would be helpful for clarity.

Reviewer 2 Report

   In the manuscript entitled “NKG2A immune checkpoint in V2δT cells: Emerging application in cancer immunotherapy”, the author discussed the current state-of-the-art knowledge related to the r2δT cells to be used in cancer immunotherapy. The significance of r2δT cells has been well illustrated in this review. Despite the potential for NKG2A immune checkpoint in V2δT cells, the following issues should be fixed before the acceptance of this review.

1)     To better emphasize the V2δT cells, the author should better compare them with other types of T cells. A table that lists the characterization of different T cells should be provided in this review.

2)     As this is a review paper, the author should briefly discuss the development of NKG2A.

3)     NKG2A is a novel immune checkpoint. Hence, the significance and advantages for this therapy should be comprehensively summarized. For example, compared with conventional immune checkpoint, what is the benefit of NKG2A?

Round 2

Reviewer 2 Report

The author has well solved the issues. Hence, this paper should be published.